# Suicidal ideation and interrelated psychiatric disturbances in rheumatoid arthritis: Evidence from a Vietnamese cohort

Ha Thi Thu Tran[1,2]☯*, Tam Minh Duong[1,2]☯, Tuan Van Nguyen[1,2], Ha Thi Thu Le[1,2], Hue Thi Doan[1,2], Yen Hoang Nguyen[1,2], Hoa Thi Nguyen[1,2], Long Thanh Nguyen[1,2], Thang Xuan Pham[1,2], An Thi Ha Tran[1,2], Phi Van Nguyen[1,2], Son Truong Hoang[1,2], Thien Cong Le[1,2], Hung Van Nguyen[3,4]

1 Department of Psychiatry, Hanoi Medical University, Hanoi, Vietnam, 2 National Institute of Mental Health, Hanoi, Vietnam, 3 Rheumatology Center, Bach Mai Hospital, Hanoi, Vietnam, 4 Internal Medicine Department, Hanoi Medical University, Hanoi, Vietnam

☯ These authors contributed equally to this work.
* tran_thuha@hmu.edu.vn

## Abstract

### Objective

This study aimed to determine the prevalence of suicidal ideation among Vietnamese patients with rheumatoid arthritis (RA), and to examine associated factors, other psychiatric disturbances, and quality of life (QoL).

### Methods

A cross-sectional study was conducted from June 2024 to December 2024 at Bach Mai Hospital, Hanoi. Eligible patients met the 1987 ACR classification criteria for RA. Suicidal ideation was assessed through structured psychiatric interviews. Depression, anxiety, insomnia, and sexual dysfunction were evaluated using the PHQ-9, HAM-A, ISI, and ASEX, respectively, while QoL was measured with the EQ-5D-5L. Correlation analyses and structural equation modeling (SEM) explored interrelationships among psychiatric disturbances. Multivariable regression was applied to identify predictors of suicidal ideation, psychiatric symptoms, and QoL.

### Results

Among 187 participants (mean age 56.9±12.7 years; 84.0% female), 13.4% reported suicidal ideation. Anxious (45.5%), insomnia (48.1%) and depressive symptoms (28.3%) were highly common, and depressive, anxiety, and insomnia symptoms were strongly interrelated (e.g., r=0.74–0.80). The mean EQ-5D-5L index value was 0.6±0.2, indicating moderate impairment in overall QoL. SEM confirmed high correlation between psychiatric (DEP, ANX) and behavioral (SLP, SEX) disturbances. Multivariable analysis showed that high disease activity, joint deformity, and

**Data availability statement:** Relevant data are within the paper and its Supporting information files. Further datasets generated and/or analysed during the current study have been deposited in the Harvard Dataverse repository and are publicly available at: https://doi.org/10.7910/DVN/KCQKHM.

**Funding:** The author(s) received no specific funding for this work.

**Competing interests:** The authors declare that no competing interests exist.

treatment-related adverse effects were consistently associated with suicidal ideation, psychiatric symptoms, and impaired QoL.

## Conclusion

Suicidal ideation and other psychiatric disturbances are commonly observedamong patients with RA in Vietnam. Disease activity, structural damage, and medication-related adverse effects were major determinants, underscoring the need for integrated mental health screening and multidisciplinary management to optimize both clinical and psychosocial outcomes.

## Introduction

Rheumatoid arthritis is a systemic autoimmune disease characterized by chronic symmetric inflammation of peripheral joints, causing progressive structural destruction and frequently accompanied by systemic manifestations. In 2019, the global prevalence of RA was estimated at 18 million cases [1]. Within the 2010 Global Burden of Disease study, disability attributable to RA ranked 42nd among 291 conditions assessed [2]. Psychiatric comorbidities are increasingly recognized in patients with RA. Depression occurs two to four times more frequently in RA patients than in the general population, and its severity correlates with pain, disease activity, and functional disability [3]. Moreover, depression in RA is associated with reduced treatment adherence, elevated healthcare costs, heightened family and social burdens [4].

Several psychiatric disturbances have been reported in RA patients, including depression [5–9], anxiety [10–12], sleep disorders [13,14], and sexual dysfunction [15,16]. However, despite the clinical significance of suicide as one of the most devastating outcomes of psychiatric illness, evidence regarding suicidal ideation in RA remains limited [17]. Suicidal ideation represents a critical and clinically severe outcome, yet most prior studies in RA focus on depression or anxiety alone, without directly evaluating suicidal thoughts or intent. The scarcity of data on suicidal ideation highlights an important gap in understanding the mental health burden of RA and limits the development of targeted interventions to prevent self-harm in this population. Additionally, the relationship between suicidal ideation, other psychiatric disturbances, and quality of life has not been systematically examined, leaving unanswered questions about how mental health and functional outcomes interact in RA. Addressing these gaps is crucial for informing early risk identification, integrating mental health assessment into rheumatology practice, and ultimately improving patient-centered outcomes.

From a psychological perspective, theoretical frameworks may help explain why patients with chronic illnesses develop suicidal thoughts. The hopelessness theory of depression proposes that persistent, uncontrollable stressors—such as chronic pain, functional decline, and fluctuating disease activity—can foster negative expectations about the future, thereby increasing vulnerability to suicidal ideation [18]. Likewise, contemporary models of chronic illness adjustment highlight the roles of perceived

burden, loss of autonomy, and disruptions in social roles and meaning-making, all of which are highly relevant to the lived experience of RA [19]. These mechanisms suggest that suicidality in RA may arise not only from depressive symptom severity but also from broader psychological and psychosocial challenges associated with chronic disease.

Given the near absence of research on suicidal ideation within RA populations, addressing this gap is essential for improving early risk identification, strengthening the integration of mental health assessment within rheumatology practice, and ultimately enhancing patient-centered outcomes. Based on this rationale, we conducted the present study with the primary aim of describing the prevalence of suicidal ideation in Vietnamese patients with RA and identifying associated clinical and sociodemographic factors. The secondary aim was to provide an integrated overview of other psychiatric disturbances—including depression, anxiety, insomnia, and sexual dysfunction—and their relationship with QoL.

## Methods

### Study design

We carried out a cross-sectional study between June 2024 and December 2024 at the Rheumatology Center and Outpatient Department of Bach Mai Hospital, enrolling patients with rheumatoid arthritis.

### Participants

Eligible participants were patients diagnosed with rheumatoid arthritis (RA) based on the 1987 American College of Rheumatology (ACR) classification criteria. These criteria required the presence of at least four features: morning stiffness lasting ≥1 hour, arthritis in three or more joint regions, arthritis of the hand joints, symmetric arthritis, rheumatoid nodules, positive rheumatoid factor, or radiographic changes characteristic of RA [20]. Individuals with cognitive impairment or hearing/speech disabilities that could impede effective communication were excluded.

### Data collection

All eligible participants were provided with detailed information regarding the study objectives, procedures, and types of data to be collected. Written informed consent was obtained prior to data collection. Demographic characteristics were recorded through structured interviews and clinical examinations, including age, sex, average income, occupation, marital status, place of residence, education level, height, and weight. Clinical information related to rheumatoid arthritis was also gathered, such as disease duration, number of hospital admissions and disease flares during the previous year, current medications and their adverse effects, duration of morning stiffness, joint deformities, as well as counts of swollen and tender joints. Duration of morning stiffness was self-reported by patients in minutes during the structured clinical interview. Patients were asked, "On average, how long does your stiffness last in the morning before it improves?" and responses were recorded in minutes. Adverse effects of RA medications were systematically recorded based on a combination of patient self-report during the structured interview and review of the patient's medical chart. Each reported or documented adverse effect was categorized according to standard clinical definitions, and the total number of adverse effects per patient was tallied for analysis. Disease activity was evaluated using the Clinical Disease Activity Index (CDAI), which integrates swollen and tender joint counts with both patient and physician global assessments, yielding a total score between 0 and 76. According to the CDAI score, patients were categorized into four groups: remission (CDAI ≤ 2.8), low disease activity (CDAI > 2.8 to ≤ 10), moderate disease activity (CDAI > 10 to ≤ 22), and high disease activity (CDAI > 22) [21].

### Study outcomes

Suicidal ideation was the primary outcome of this study. It was assessed through direct clinical evaluation conducted by board-certified psychiatrists during structured interviews. The assessment focused on the presence of suicidal thoughts, intent, or planning, based on established psychiatric diagnostic guidelines [22,23]. Patients were classified into two

categories: absence of suicidal ideation and presence of suicidal ideation. For the purpose of this study, any report of suicidal thoughts, regardless of frequency or intensity, was considered a positive outcome.

Mental disturbances—including anxiety, depression, insomnia, and sexual dysfunction—were secondary outcomes in this study. Because the measurement instruments for these secondary outcomes (PHQ-9, HAM-A, ISI, ASEX, and EQ-5D-5L) were not the primary focus of the research, only a brief assessment of their internal consistency and basic construct validity was performed. All instruments had previously been translated using standardized forward–backward translation and harmonization procedures, followed by pilot testing and face-validity evaluation to ensure clarity, semantic equivalence, and cultural appropriateness. Cronbach's alpha was calculated for all scales, and confirmatory factor analysis (CFA) was performed when appropriate. Overall, PHQ-9, HAM-A and ISI showed high internal consistency and acceptable item–item and item–total correlation patterns, with CFA models demonstrating borderline but interpretable fit. The ASEX scale also showed good internal consistency; however, factor-analytic evaluation was limited by the small number of participants who reported recent sexual activity. For EQ-5D-5L, internal consistency was moderate and item–score correlations aligned with its conceptual domains; factorial validation was not applicable because the instrument is formative rather than reflective (S1 Table).

Depression was assessed using the Patient Health Questionnaire-9 (PHQ-9), a 9-item self-report measure with total scores ranging from 0 to 27, where higher scores indicate more severe depressive symptoms. In this study, a total score ≥ 8 was considered indicative of clinically relevant depressive symptoms [24]. Anxiety was evaluated using the Hamilton Anxiety Rating Scale (HAM-A), comprising 14 items covering psychological and somatic symptoms. Each item is scored from 0 to 4, yielding a total score of 0–56. A total score >13 was used to define clinically significant anxiety [25]. Sleep quality was assessed with the Insomnia Severity Index (ISI), a 7-item self-report scale (score range 0–28), with higher scores indicating more severe insomnia. Standard cut-offs were applied to classify insomnia severity [26]. The ISI had been validated in multiple studies, demonstrating high accuracy and reliability [26–28]. Sexual dysfunction was measured using the Arizona Sexual Experiences Scale (ASEX), which evaluated sexual drive, arousal, physiological ability, orgasm, and satisfaction. Each item is scored 1–6; a total score ≥19 indicated sexual dysfunction [29].

Quality of life was assessed using the European Quality of Life 5 Dimensions 5 Level (EQ-5D-5L), which comprises five domains: mobility, self-care, usual activities, pain/discomfort, and anxiety/depression. Each domain was rated across five levels, ranging from 1 (no problems) to 5 (extreme problems). The five domains were subsequently converted into an index score using value sets derived from a previously validated study conducted in the Vietnamese population [30]. Previous research had demonstrated that the EQ-5D-5L possesses strong validity and reliability across various populations [31].

## Sample size and sampling

Sample size was calculated based on the prevalence of prevalence of suicidal ideation among patients with RA in a previous study [17]. Assuming a prevalence of suicidal ideation of 30% among patients with RA, a margin of error of 8%, and a confidence level of 95%, the minimum sample size was 151 patients. Accounting for the non-participate rate of 20%, we planned to recruit 180 participants.

Consecutive sampling was employed, enrolling all eligible patients who presented to the study sites during the data collection period.

## Statistical analysis

Categorical variables were summarized as frequencies and percentages, whereas continuous variables were presented as means with standard deviations or medians with interquartile ranges (Q1–Q3). Stacked bar charts were employed to visualize the distribution of responses on the psychological assessment scales. Correlations between these scales were examined using Spearman's rank correlation and illustrated with scatter plots complemented by LOESS curves to depict linear trends.

We used structural equation modeling (SEM) to examine the latent structure and interrelationships among psychiatric and behavioral disturbances because SEM permitted (1) representation of underlying latent constructs (e.g., psychiatric and behavioral factors) from multiple observed indicators while explicitly modelling measurement error, and (2) simultaneous estimation of associations between latent factors — capabilities that simple correlation or multiple regression frameworks do not provide. SEM therefore allows us to test whether psychiatric symptoms cluster together as coherent latent constructs and to quantify the degree of overlap between psychological (depression, anxiety) and behavioral (insomnia, sexual dysfunction) disturbances in a single integrated model. We specified a two-factor measurement model with latent factors PSY (depression, anxiety) and BEHAV (insomnia, sexual dysfunction), and estimated the covariance between them. The model was estimated using robust maximum likelihood (MLR) to account for modest non-normality of item distributions, and missing data were handled with full information maximum likelihood (FIML). Model fit was evaluated using standard indices — chi-square ($\chi^2$), Comparative Fit Index (CFI), Tucker–Lewis Index (TLI), Root Mean Square Error of Approximation (RMSEA) with 90% CI, and Standardized Root Mean Square Residual (SRMR). We pre-specified conventional cutoffs for acceptable fit (CFI/TLI ≥ 0.90 acceptable, ≥ 0.95 good; RMSEA ≤0.08 acceptable, ≤ 0.06 good; SRMR ≤0.08). Parameter estimates were interpreted alongside these fit indices. Prior to SEM we inspected univariate skewness and kurtosis for observed indicators and examined bivariate correlations for signs of multicollinearity. Because several indicators exhibited modest skewness, we used the robust estimator (MLR). Multicollinearity among observed variables was assessed (variance inflation factors and pairwise correlations) and did not indicate problematic collinearity. Sample size (n = 187) was judged adequate for the simple two-factor model (rule-of-thumb ≥10 observations per estimated parameter). Detailed diagnostics (skewness/kurtosis, VIFs) and full fit statistics are provided in S2 Table.

Binary logistic regression was used to identify factors associated with the presence of suicidal ideation, depression, anxiety, insomnia, and sexual dysfunction. Linear regression was performed to explore factors related to quality of life, as measured by the EQ-5D-5L. Candidate variables for multivariable regression models were selected through a rigorous process: (1) consideration of risk factors previously documented in the literature on depression among patients with rheumatoid arthritis; (2) evaluation of clinical plausibility to ensure practical relevance of the models; and (3) preliminary screening through univariable analyses.

All analyses were conducted using R version 4.3.2, with statistical significance defined as a two-sided P-value less than 0.05.

## Ethical considerations

The study was conducted in accordance with the Declaration of Helsinki and approved by the Institutional Review Board of Hanoi Medical University under decision No. 965/GCN-HĐĐĐNCYSSH-ĐHYHN, dated November 29, 2023. Participants provided written informed consent. Their responses were anonymous and kept confidential.

## Results

A total of 187 participants were included (mean age 56.9 ± 12.7 years; 84.0% female). Most patients had RA for over one year (98.4%), with moderate or high disease activity in 78.6%, and joint deformities observed in 43.3%. Adverse drug reactions occurred in 21.4% of patients (Table 1).

Suicidal ideation was reported in 13.4% of participants. Psychiatric disturbances were common: 45.5% had anxiety, 48.1% had insomnia, 28.3% had depression and 8.6% reported sexual dysfunction. The mean EQ-5D-5L index was 0.6 ± 0.2, indicating moderate impairment in quality of life (Table 2).

Correlation analyses revealed strong associations between depression severity and both anxiety (r = 0.74, p < 0.001) and insomnia (r = 0.80, p < 0.001), while sexual dysfunction showed only weak correlation with depression (r = 0.27, p = 0.007) (Fig 1). SEM confirmed that depression (DEP) and anxiety (ANX) loaded strongly onto the latent factor PSY, whereas sexual dysfunction (SEX) and insomnia (SLP) loaded onto BEHAV, with a high correlation between PSY and BEHAV, reflecting substantial overlap between psychological and behavioral disturbances (Fig 2).

**Table 1. Participant's characteristics (n = 187).**

| Characteristic | Results |
|---|---|
| Age (years), Mean±SD | 56.9±12.7 |
| Gender female, n (%) | 157 (84.0%) |
| Inpatient/Outpatient, n (%) | |
| Inpatient | 140 (74.9%) |
| Outpatient | 47 (25.1%) |
| Location of residence, n (%) | |
| Urban | 65 (34.8%) |
| Rural | 122 (65.2%) |
| Educational level, n (%) | |
| High school and below | 106 (56.7%) |
| College and above | 81 (43.3%) |
| Marital status, n (%) | |
| Single/Divorced/Widow | 17 (9.1%) |
| Married | 170 (90.9%) |
| Occupational, n (%) | |
| Worker | 12 (6.4%) |
| Farmer | 62 (33.2%) |
| Trader | 18 (9.6%) |
| Government employee | 10 (5.3%) |
| Housewife | 13 (7.0%) |
| Retired | 39 (20.9%) |
| Student | 4 (2.1%) |
| Self-employed | 21 (11.2%) |
| Unemployed | 2 (1.1%) |
| Others | 6 (3.2%) |
| Individual income, n (%) | |
| Below minimum wage | 110 (58.8%) |
| Above minimum wage | 77 (41.2%) |
| Had medical insurance, n (%) | 147 (78.6%) |
| BMI group, n (%) | |
| Underweight | 31 (16.6%) |
| Normal | 121 (64.7%) |
| Overweight | 35 (18.7%) |
| Physical conditions, n (%) | 104 (55.6%) |
| Mental conditions, n (%) | 3 (1.6%) |
| Disease duration ≥ 1 year, n (%) | 184 (98.4%) |
| Hospitalization due to RA in the past year, n (%) | 108 (57.8%) |
| Had RA flare in the past year, n (%) | 91 (48.7%) |
| Morning stiffness duration (minutes), Mean±SD | 33.5±26.5 |
| Joint deformity, n (%) | 81 (43.3%) |
| Disease activity, n (%) | |
| Remission | 5 (2.7%) |
| Low disease activity | 35 (18.7%) |
| Moderate disease activity | 52 (27.8%) |
| High disease activity | 95 (50.8%) |

*(Continued)*

**Table 1.** (Continued)

| Characteristic | Results |
|---|---|
| Medication used, n (%) | |
| NSAIDs | 89 (47.6%) |
| Corticosteroids | 139 (74.3%) |
| DMARDs | 100 (53.5%) |
| Biological medications | 52 (27.8%) |
| Other medications | 50 (26.7%) |
| Medication's AEs, n (%) | 40 (21.4%) |
| Adrenal insufficiency | 18 (9.6%) |
| Gastritis/Peptic ulcer | 11 (5.9%) |
| Cushing's syndrome | 9 (4.8%) |
| Osteoporosis | 2 (1.1%) |
| Peripheral edema | 1 (0.5%) |
| GI bleeding | 1 (0.5%) |

Abbreviations: SD, standard deviation; RA, rheumatoid arthritis; NSAIDs, nonsteroidal anti-inflammatory drugs; DMARDs, disease-modifying antirheumatic drugs; BMI, body mass index; GI, gastrointestinal.

High disease activity emerged as the most consistent predictor, demonstrating strong associations across nearly all outcomes, including suicidal ideation (OR 11.2, 95% CI: 3.07–57.9, p < 0.001), depression (OR 5.05, 95% CI: 2.26–12.1, p < 0.001), anxiety (OR 3.80, 95% CI: 1.88–8.00, p < 0.001), insomnia (OR 4.74, 95% CI: 2.30–10.2, p < 0.001), and impaired QoL (β = −0.13, 95% CI: −0.18 to −0.07, p < 0.001). Joint deformity was also significantly related to suicidal ideation (OR 2.96, 95% CI: 1.05–8.93, p = 0.041) and poorer QoL (β = −0.06, 95% CI: −0.11 to −0.01, p = 0.018). AEs of medication correlated with suicidal ideation (OR 3.93, 95% CI: 1.29–12.2, p = 0.016), as well as depression, anxiety, and insomnia. Female gender was associated with higher risks of depression (OR 3.62, 95% CI: 1.06–17.3, p = 0.044) and anxiety (OR 3.82, 95% CI: 1.43–11.7, p = 0.008). In contrast, sociodemographic factors such as education, income, and marital status showed weaker and less consistent relationships (Table 3).

## Discussion

The primary aim of this study was to determine the prevalence of suicidal ideation among Vietnamese patients with rheumatoid arthritis. We found that 13.4% of participants reported suicidal thoughts. This proportion is lower than the 31.9% prevalence reported in a 2022 study of 72 RA outpatients in Mexico, which also noted an 8.3% rate of suicide attempts when assessed using a self-report suicidality scale [17]. Findings from other autoimmune and chronic inflammatory diseases provide additional context. A 2013 study from Hong Kong involving 367 systemic lupus erythematosus (SLE) patients aged 18–65 years reported a 12% prevalence of suicidal ideation using direct clinical questioning [32]. A 2025 meta-analysis including 1,515 SLE patients from multiple regions estimated suicidal ideation at approximately 16.8%, with suicide attempts and suicide deaths reported in 2.7% and 0.9% of cases, respectively [33]. In fibromyalgia, a meta-analysis published in 2019 that pooled data from 394,087 patients across North America, Europe, and Asia identified a suicidal ideation prevalence of nearly 30% and a pooled suicide-attempt prevalence of about 6% [34]. In multiple sclerosis (MS), a 2020 study of 162 adults in Central Europe reported elevated rates of suicidal ideation compared with the general population [35], while a 2024 Iranian study of 400 MS outpatients identified a lifetime suicidal ideation prevalence of 37.2% [36]. Several methodological and clinical differences across studies may explain the variability in reported rates, including differences in sample size, disease severity, patient demographics, healthcare setting (community vs. tertiary hospitals), cultural context, and—most importantly—variability in suicidality assessment methods (e.g., structured

**Table 2. Suicidal ideation, quality of life, and other psychiatric disturbances of the participants (n = 187).**

| Characteristic | Results |
|---|---|
| Suicidal ideation | 25 (13.4%) |
| Depression levels according to PHQ-9 classification, n (%) | |
| None | 104 (55.6%) |
| Mild | 46 (24.6%) |
| Moderate | 25 (13.4%) |
| Severe | 11 (5.9%) |
| Very severe | 1 (0.5%) |
| Had depression, n (%) | 53 (28.3%) |
| Anxiety levels according to Hamilton classification, n (%) | |
| None | 102 (54.5%) |
| Mild | 41 (21.9%) |
| Moderate | 23 (12.3%) |
| Severe | 13 (7.0%) |
| Very severe | 8 (4.3%) |
| Had anxiety, n (%) | 85 (45.5%) |
| Insomnia levels according to ISI classification, n (%) | |
| None | 97 (51.9%) |
| Mild | 57 (30.5%) |
| Moderate | 31 (16.6%) |
| Severe | 2 (1.1%) |
| Had insomnia, n (%) | 90 (48.1%) |
| Had sexual dysfunction, n (%) | 16 (8.6%) |
| EQ-5D-5L index value, MEAN±SD | 0.6 ± 0.2 |

Abbreviations: PHQ-9, Patient Health Questionnaire 9; ISI, Insomnia Severity Index; EQ-5D-5L, European Quality of Life 5 Dimensions 5 Level; SD, standard deviation.

interviews, direct questions, or self-report scales). Against this wider background, the relatively high prevalence of suicidal ideation observed in our Vietnamese RA cohort adds important evidence to an area where data remain sparse and further reinforces the substantial burden of depressive symptoms among individuals with RA. Suicidal ideation represents one of the most serious manifestations of depression, frequently associated with persistent low mood, hopelessness, uncontrolled chronic pain, and social isolation—factors that are highly prevalent in this population. Taken together, these findings emphasize that suicidal ideation is a critical but often underrecognized issue in RA care, particularly when clinicians lack sufficient training to identify atypical depressive symptoms. The relatively high prevalence documented in our study underscores the urgent need to integrate routine mental health screening into rheumatology practice, with the goal of enabling early detection of high-risk mood disturbances, improving adherence to long-term treatment, enhancing overall quality of life, and ultimately reducing preventable mortality in patients living with RA. It is important to clarify that suicidal ideation, as assessed in this study, represents a marker of psychological distress and potential suicide risk, but does not equate to suicide attempts or suicide mortality. The presence of suicidal thoughts indicates elevated vulnerability and the need for clinical attention, rather than definitive suicidal behavior.

Our findings also demonstrated the frequent presence of other psychiatric disturbances, consistent with previous studies that have reported high prevalence of depression [5–9], anxiety [10–12], insomnia [13,14], and sexual dysfunction

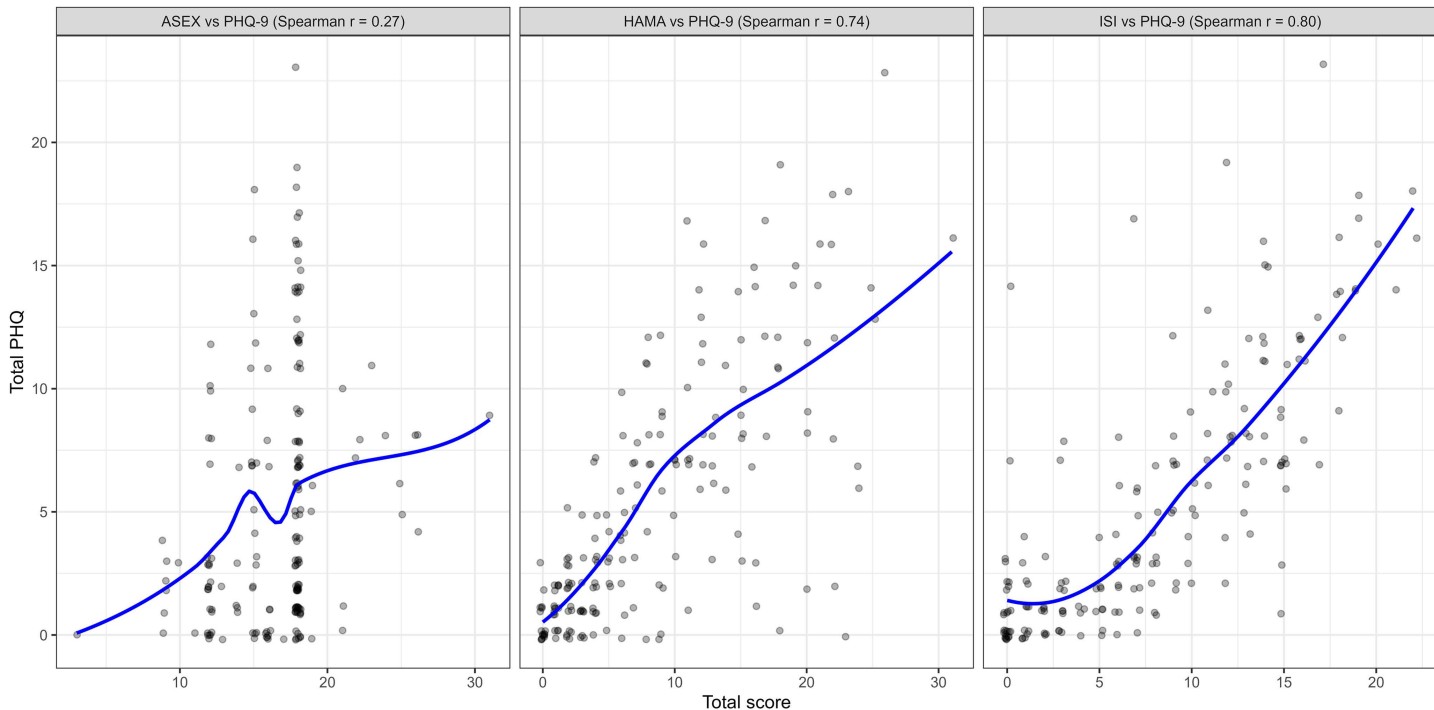

**Fig 1. Associations among psychiatric disturbances of the participants (n = 187).**

[15,16] among patients with RA. Biologically, these disturbances are interconnected through shared pathways involving systemic inflammation, neuroendocrine dysregulation, and chronic pain sensitization, which together sustain mood and sleep disturbances in RA [37–40]. Socially, factors such as functional disability, occupational loss, dependence on caregivers, and reduced social participation further exacerbate psychological vulnerability, creating a complex interplay between disease burden and mental health [41,42]. Clinically, their presence contributes not only to impaired quality of life but also to poorer treatment adherence, reduced functional recovery, and worse long-term outcomes [43–46]. Importantly, evidence from our study demonstrated strong correlations among depression, anxiety, insomnia, and sexual dysfunction in both Spearman analyses and SEM modeling. This suggests that the occurrence of one disturbance substantially increases the likelihood of others, reinforcing the view that psychological morbidity in RA is multidimensional rather than isolated. These findings strongly support the need for comprehensive mental health screening and integrated management strategies, ensuring that co-occurring conditions are identified early and addressed in a coordinated manner to optimize both psychological well-being and disease control.

Building on these interrelationships, we next examined clinical determinants that may underlie this pattern of psychological morbidity. High disease activity emerged in our study as the most consistent predictor of adverse psychological and quality-of-life outcomes, showing strong associations with suicidal ideation, depression, anxiety, insomnia, and impaired QoL. To our knowledge, no previous study has directly examined risk factors for suicidal ideation in RA, yet prior evidence has consistently demonstrated a close link between higher disease activity and greater severity of depressive symptoms [43,47–50]. Given the well-established association between depression and suicidal thoughts, it is reasonable to infer that uncontrolled disease activity may indirectly increase suicide risk through its impact on mood. In addition, existing studies have reported significant relationships between disease activity and anxiety [51], sleep disturbance [52,53], sexual dysfunction [54,55], and diminished quality of life [56–58], further supporting our findings. Biologically, the underlying

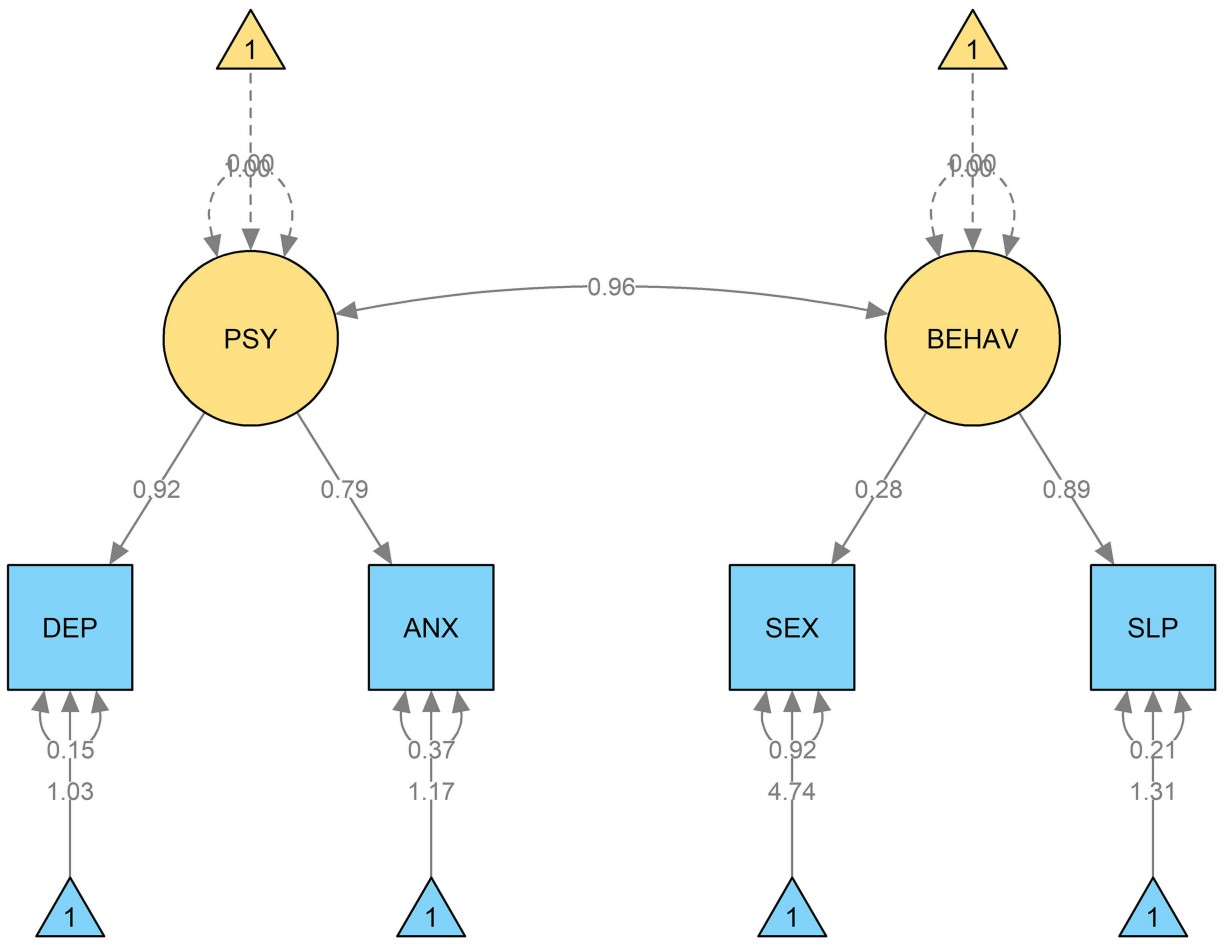

**Fig 2. SEM model illustrating the correlations among psychiatric disturbances of the participants.** Abbreviations: DEP, Depression (PHQ-9); ANX, Anxiety (HAMA); SEX, Sexual dysfunction (ASEX); SLP, Insomnia (ISI); PSY, Latent factor representing psychiatric disturbances; BEHAV, Latent factor representing behavioral disturbances.

mechanisms likely involve chronic systemic inflammation, heightened pain sensitivity, and neuroendocrine dysregulation, which contribute to both disease progression and psychological morbidity [37–40].

Beyond disease activity, treatment-related factors also played an important role. Our study demonstrated that adverse effects related to pharmacological treatment were significantly associated with suicidal ideation, as well as with depression, anxiety, and insomnia. There is substantial evidence linking treatment-related complications with increased psychological distress. Previous studies have reported that corticosteroid-related side effects, such as weight gain, Cushingoid features, or sleep disturbance, are strongly associated with depressive symptoms [59,60], while gastrointestinal and musculoskeletal complications of NSAIDs or DMARDs may contribute to chronic anxiety and insomnia [61–63]. Moreover, the burden of persistent or severe AEs often leads to treatment discontinuation or reduced adherence, further exacerbating disease activity and indirectly worsening mental health outcomes. The mechanisms underlying these associations are likely multifactorial, including direct neuropsychiatric effects of certain agents (e.g., corticosteroid-induced mood disorders) [59,60], systemic inflammation triggered by drug toxicity [62,64], and psychosocial consequences of living with long-term treatment side effects [65,66]. It should be noted that some odds ratios, particularly for suicidal ideation (e.g., OR 11.2, 95% CI 3.07–57.9), are large with wide confidence intervals. This is likely due to the limited number of events (n = 25

**Table 3. Multivariable analysis of factors associated with suicidal ideation, quality of life, and psychiatric disturbances in the study population (n = 187).**

| Characteristic | Suicidal ideation | | Depression | | Anxiety | | Insomnia | | Sexual dysfunction | | QoL | |
|---|---|---|---|---|---|---|---|---|---|---|---|---|
| | OR | 95% CI | OR | 95% CI | OR | 95% CI | OR | 95% CI | OR | 95% CI | Beta | 95% CI |
| Age (years) | 1.00 | 0.95, 1.05 | 1.01 | 0.98, 1.05 | 1.01 | 0.99, 1.04 | 1.01 | 0.98, 1.04 | 0.87 | 0.81, 0.93 | 0.00 | 0.00, 0.00 |
| Gender female | 5.85 | 0.89, 121 | 3.62 | 1.06, 17.3 | 3.82 | 1.43, 11.7 | 2.84 | 1.07, 8.37 | 5.63 | 0.76, 126 | −0.01 | −0.08, 0.06 |
| Educational level: College or above | 0.62 | 0.16, 2.15 | 1.03 | 0.44, 2.45 | 1.07 | 0.50, 2.33 | 0.57 | 0.26, 1.25 | 0.62 | 0.12, 2.75 | 0.01 | −0.05, 0.07 |
| Income above minimum wage | 1.29 | 0.39, 4.22 | 0.50 | 0.20, 1.17 | 0.89 | 0.42, 1.91 | 0.82 | 0.37, 1.82 | 3.31 | 0.77, 15.9 | 0.06 | 0.00, 0.12 |
| Married | 0.70 | 0.15, 3.86 | 0.69 | 0.20, 2.50 | 0.71 | 0.20, 2.40 | 0.41 | 0.10, 1.49 | 3.84 | 0.37, 105 | 0.01 | −0.09, 0.10 |
| Hospitalization due to RA | 1.03 | 0.32, 3.37 | 1.97 | 0.85, 4.73 | 1.36 | 0.66, 2.85 | 2.30 | 1.07, 5.12 | 1.67 | 0.39, 7.91 | −0.02 | −0.07, 0.04 |
| Had RA flare in the past year | 0.50 | 0.14, 1.62 | 0.63 | 0.27, 1.41 | 1.08 | 0.53, 2.18 | 1.23 | 0.59, 2.59 | 0.65 | 0.14, 2.64 | −0.06 | −0.11, 0.00 |
| Joint deformity | 2.96 | 1.05, 8.93 | 1.68 | 0.80, 3.56 | 1.80 | 0.92, 3.58 | 0.78 | 0.38, 1.58 | 1.25 | 0.29, 5.20 | −0.06 | −0.11, −0.01 |
| High disease activity | 11.2 | 3.07, 57.9 | 5.05 | 2.26, 12.1 | 3.80 | 1.88, 8.00 | 4.74 | 2.30, 10.2 | 6.85 | 1.21, 55.9 | −0.13 | −0.18, −0.07 |
| Had medication's AEs | 3.93 | 1.29, 12.2 | 3.19 | 1.37, 7.60 | 2.47 | 1.11, 5.70 | 4.92 | 2.08, 12.6 | 1.52 | 0.28, 6.87 | −0.07 | −0.13, 0.00 |

Abbreviations: RA, rheumatoid arthritis; QoL, quality of life; OR, odds ratio; CI, confidence interval; AEs, adverse effects.

participants reporting suicidal ideation) and may reflect potential overfitting of the regression model. These estimates should therefore be interpreted primarily as indicators of statistical association rather than as precise quantification of risk, and with appropriate caution given the uncertainty inherent in analyses of relatively rare outcomes.

Our findings indicate that both uncontrolled disease activity and treatment-related adverse effects function as key drivers of psychological morbidity in RA. These two factors not only contribute directly to the development of depression, anxiety, insomnia, and suicidal ideation, but also interact with each other to create a vicious cycle in which uncontrolled inflammation increases the risk of complications, while drug-related toxicity further undermines adherence and worsens disease outcomes. This interplay highlights the need for an integrated management strategy that simultaneously prioritizes tight control of disease activity, early identification and mitigation of medication-related AEs, and systematic mental health screening. Such approach is essential to improve treatment adherence, preserve functional capacity, enhance quality of life, and ultimately reduce the overall burden of RA.

This study has several methodological limitations that should be considered when interpreting and generalizing the findings. First, as random sampling was not applied, the study population may not fully represent the broader RA community. Most participants in our cohort had high disease activity and long disease duration, reflecting a more severe subgroup of patients who typically seek care at tertiary centers. This selection bias may have led to overestimation of the prevalence of suicidal ideation, impaired QoL, and other psychiatric disturbances, and may also have influenced the observed associations between clinical characteristics and mental health outcomes. Second, the cross-sectional design limits our ability to determine the temporal course or causal relationships between disease activity, treatment-related factors, and psychological outcomes. Suicidal ideation and psychiatric symptoms in chronic illnesses such as RA are dynamic processes that fluctuate over time depending on disease control, personal life events, and coping strategies. Assessing patients at a single time point does not capture the complexity and variability of these conditions, and restricts our capacity to evaluate the predictive value of identified risk factors. Consequently, while we observed strong associations—such as between high disease activity and suicidal ideation or between adverse drug events and psychological outcomes—these should be interpreted as correlations rather than evidence of direct causality. Bidirectional effects are possible; for example, pre-existing depression or anxiety may influence patients' perceptions of disease severity or treatment adherence, and drug-related complications may both result from and contribute to psychological distress. Longitudinal studies are therefore needed to clarify causal pathways and to examine how mental health

outcomes evolve across the disease trajectory in RA. Thirdly, cultural and health system factors in Vietnam may have influenced the reporting of psychological symptoms. Mental health stigma and social norms emphasizing resilience could discourage patients from openly disclosing depressive symptoms, anxiety, or suicidal thoughts. Consequently, self-reported or interview-based assessments may be affected by social desirability bias, potentially leading to under-estimation of the prevalence and severity of psychiatric disturbances. Limited integration of mental health services within rheumatology care may further reduce opportunities for disclosure, highlighting the importance of culturally sensi-tive and systematic screening approaches in future studies. Additionally, suicidal ideation in this study was assessed through structured clinician interviews conducted by board-certified psychiatrists, without the use of a standardized instrument, this approach may limit comparability with international studies and introduce potential rater-dependent variability. Moreover, suicidal ideation was operationalized as a binary outcome encompassing any report of suicidal thoughts, intent, or planning. This dichotomization may have obscured clinically meaningful distinctions between passive death wishes, active suicidal thoughts, and suicidal intent or planning, which carry different prognostic and clinical implications. It should also be acknowledged that some degree of conceptual overlap between depressive symptoms and suicidal ideation is unavoidable, as suicidality-related content is embedded in commonly used depres-sion measures such as the PHQ-9. Nevertheless, suicidal ideation in this study was assessed independently through clinician-administered interviews, allowing it to be examined as a distinct clinical outcome rather than merely a compo-nent of self-reported depressive severity. Lastly, we did not assess potentially influential cognitive or psychosocial fac-tors—such as coping styles, perceived social support, or self-efficacy—that may mediate or moderate the relationships between disease activity and psychological outcomes. Incorporating these dimensions in future research would help clarify underlying mechanisms and strengthen the explanatory value of psychological models in RA.

## Conclusion

This study demonstrates that suicidal ideation and other psychiatric disturbances are commonly observedamong Vietnam-ese patients with RA. High disease activity, joint deformity, and treatment-related adverse effects were major predictors, underscoring the interplay between disease burden and mental health. These findings highlight the need for integrated mental health screening and multidisciplinary care in routine rheumatology practice to improve both clinical outcomes and quality of life. In routine rheumatology settings, integrated care may involve brief screening questions for mood, sleep, and suicidal thoughts during regular visits, followed by referral to mental health professionals when clinically indicated.

## Supporting information

**S1 Table. Preliminary validation metrics of psychometric scales in the study cohort.**
(DOCX)

**S2 Table. Structural equation modeling (SEM) fit indices, standardized loadings, and diagnostics.**
(DOCX)

**S1 Fig. Depression profile based on PHQ-9 scale (n = 187).**
(TIFF)

**S2 Fig. Anxiety profile based on HAM-A scale (n = 187).**
(TIFF)

**S3 Fig. Sleep disturbance profile based on the ISI scale (n = 187).**
(TIFF)

**S4 Fig. Sexual dysfunction profile based on the ASEX scale (n = 187).**
(TIFF)

**S5 Fig. Quality of life profile based on the EQ-5D-5L scale (n = 187).**
(TIFF)

**S1 File. rasuicide_dataset.**
(XLSX)

**S2 File. rasuicide_output.**
(RMD)

## Author contributions

**Conceptualization:** Ha Thi Thu Tran, Tam Minh Duong, Hung Van Nguyen, Ha Thi Thu Le, Tuan Van Nguyen.

**Data curation:** Ha Thi Thu Tran.

**Formal analysis:** Ha Thi Thu Tran.

**Funding acquisition:** Ha Thi Thu Tran.

**Investigation:** Ha Thi Thu Tran, Ha Thi Thu Le, Hue Thi Doan, Yen Hoang Nguyen, Hoa Thi Nguyen, Long Thanh Nguyen, Thang Xuan Pham, An Thi Ha Tran, Phi Van Nguyen, Son Truong Hoang, Thien Cong Le.

**Methodology:** Ha Thi Thu Tran, Tam Minh Duong, Hung Van Nguyen, Ha Thi Thu Le, Hue Thi Doan, Yen Hoang Nguyen, Hoa Thi Nguyen, Long Thanh Nguyen, Thang Xuan Pham, An Thi Ha Tran, Phi Van Nguyen, Son Truong Hoang, Thien Cong Le, Tuan Van Nguyen.

**Project administration:** Ha Thi Thu Tran, Tam Minh Duong, Hung Van Nguyen, Tuan Van Nguyen.

**Resources:** Ha Thi Thu Tran, Tuan Van Nguyen.

**Software:** Ha Thi Thu Tran.

**Supervision:** Ha Thi Thu Tran, Tam Minh Duong, Hung Van Nguyen, Tuan Van Nguyen.

**Validation:** Ha Thi Thu Tran, Hung Van Nguyen, Tuan Van Nguyen.

**Visualization:** Ha Thi Thu Tran.

**Writing – original draft:** Ha Thi Thu Tran.

**Writing – review & editing:** Ha Thi Thu Tran, Tam Minh Duong, Hung Van Nguyen, Ha Thi Thu Le, Hue Thi Doan, Yen Hoang Nguyen, Hoa Thi Nguyen, Long Thanh Nguyen, Thang Xuan Pham, An Thi Ha Tran, Phi Van Nguyen, Son Truong Hoang, Thien Cong Le, Tuan Van Nguyen.

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
