## [Decision Letter · Decision Letter 0]

20 Jan 2026

Dear Dr. Thi Thu Tran,

Thank you for submitting your manuscript to PLOS ONE. After careful consideration, we feel that it has merit but does not fully meet PLOS ONE’s publication criteria as it currently stands. Therefore, we invite you to submit a revised version of the manuscript that addresses the points raised during the review process.

**ACADEMIC EDITOR: Major Revision**

We look forward to receiving your revised manuscript.

Kind regards,

Marwan Salih Al-Nimer, MD, PhD

Academic Editor

PLOS One

3. We notice that your Figure S2 is uploaded with the file type 'Figure'. Please amend the file type to 'Supporting Information'. Please ensure that each Supporting Information file has a legend listed in the manuscript after the references list.

Additional Editor Comments:

1-Add the value of Cronbach's alpha

2- Type the references according to the PLoS ONE

3- Improve the quality of the figures

Reviewers' comments:

Reviewer's Responses to Questions

**Comments to the Author**

1. Is the manuscript technically sound, and do the data support the conclusions?

Reviewer #1: Yes

Reviewer #2: Yes

2. Has the statistical analysis been performed appropriately and rigorously?

Reviewer #1: Yes

Reviewer #2: Yes

3. Have the authors made all data underlying the findings in their manuscript fully available?

Reviewer #1: Yes

Reviewer #2: Yes

4. Is the manuscript presented in an intelligible fashion and written in standard English?

Reviewer #1: Yes

Reviewer #2: Yes

Reviewer #1: This manuscript addresses an important and underexplored topic: suicidal ideation among patients with rheumatoid arthritis. The study is generally well conducted, clearly written, and the revisions have substantially improved methodological transparency. From a psychiatric perspective, the manuscript is suitable for publication, with several points that could further strengthen conceptual clarity and clinical interpretation.

Major comments

Title clarity

The term “mental health disturbances” in the title is relatively broad and nonspecific. Given that the study explicitly examines depression, anxiety, insomnia, sexual dysfunction, and suicidal ideation, the authors may consider revising the title to reflect these specific psychiatric domains more clearly. This would improve conceptual precision and better align the title with the actual content of the manuscript.

Clarification of suicidal ideation assessment

Suicidal ideation was assessed through structured psychiatric interviews, which is a strength. However, the manuscript treats suicidal ideation as a binary outcome without clarifying its clinical scope. Please briefly clarify whether suicidal ideation primarily reflected passive death wishes, active suicidal thoughts, or any planning or intent, and acknowledge in the Discussion that different forms of ideation carry different clinical implications.

Conceptual overlap with depressive symptoms

Because depressive symptom measures (e.g., PHQ-9) include suicidality-related content, some conceptual overlap between depression and suicidal ideation is unavoidable. Please add a brief acknowledgment of this issue and emphasize that suicidal ideation was assessed independently via clinician interview rather than self-report alone.

Interpretation of large effect sizes

Some odds ratios for suicidal ideation are large with wide confidence intervals due to the limited number of events. It would be helpful to further emphasize that these estimates should be interpreted as indicators of association rather than precise risk quantification.

Minor comments

Terminology

Please clarify that suicidal ideation represents a marker of psychological distress and potential risk, but does not equate to suicide attempts or suicide mortality.

Clinical implications

The conclusion appropriately calls for integrated mental health screening. Adding one brief, concrete example (e.g., routine screening questions or referral pathways) may further enhance clinical relevance for non-psychiatric readers.

Reviewer #2: Overall Assessment

This is an excellent and timely study that is highly relevant to contemporary Primary Health Care. The manuscript is logically structured, demonstrating high standards of academic writing and a clear understanding of the subject matter.

Section-by-Section Analysis

Introduction: The theoretical framework is sound. It effectively contextualizes the research problem and establishes a compelling rationale for the study.

Methodology: The research design is articulated with precision. The methodology is well-explained, providing sufficient detail for reproducibility and justifying the chosen approach.

Discussion: The interpretation of the findings is generally well-executed, situating the results within the broader context of existing Primary Health Care research.

Limitations: The authors have shown commendable reflexivity. The study limitations are explicitly "enlightened" and transparently discussed, which strengthens the integrity of the findings.

**Do you want your identity to be public for this peer review?** For information about this choice, including consent withdrawal, please see our Privacy Policy

Reviewer #1: No

Reviewer #2: **Yes:** Roland Mbuyi Tshibeya

---

## [Author Response · Author response to Decision Letter 1]

26 Jan 2026

Responses to Comments

Reviewer 1:

Comment 1:

The term “mental health disturbances” in the title is relatively broad and nonspecific. Given that the study explicitly examines depression, anxiety, insomnia, sexual dysfunction, and suicidal ideation, the authors may consider revising the title to reflect these specific psychiatric domains more clearly. This would improve conceptual precision and better align the title with the actual content of the manuscript.

Response:

We agree that the term “mental health disturbances” may be overly broad, and we have revised the title to more explicitly reflect the specific psychiatric domains examined in this study. The updated title improves conceptual clarity and better aligns with the manuscript’s content.

Comment 2:

Suicidal ideation was assessed through structured psychiatric interviews, which is a strength. However, the manuscript treats suicidal ideation as a binary outcome without clarifying its clinical scope. Please briefly clarify whether suicidal ideation primarily reflected passive death wishes, active suicidal thoughts, or any planning or intent, and acknowledge in the Discussion that different forms of ideation carry different clinical implications.

Response:

We would like to clarify that the clinical scope of suicidal ideation was explicitly defined in the Methods section (Study outcomes). Suicidal ideation was assessed through structured clinical interviews conducted by board-certified psychiatrists and encompassed the presence of suicidal thoughts, intent, or planning, in accordance with established psychiatric diagnostic guidelines. For the purposes of analysis, any report of suicidal thoughts—regardless of frequency or intensity—was classified as a positive outcome, and the variable was therefore treated as binary.

We agree with the reviewer that dichotomizing suicidal ideation may obscure clinically meaningful distinctions between passive death wishes, active suicidal thoughts, and suicidal intent or planning, each of which carries different prognostic and clinical implications. Accordingly, we have added a statement in the Discussion acknowledging this limitation.

Comment 3:

Because depressive symptom measures (e.g., PHQ-9) include suicidality-related content, some conceptual overlap between depression and suicidal ideation is unavoidable. Please add a brief acknowledgment of this issue and emphasize that suicidal ideation was assessed independently via clinician interview rather than self-report alone.

Response:

We agree that some degree of overlap between depressive symptom measures and suicidal ideation is unavoidable. This point has now been explicitly acknowledged in the Discussion section, where we note the potential conceptual overlap while emphasizing that suicidal ideation in this study was assessed independently through clinician-administered structured interviews rather than self-report measures alone.

Comment 4:

Some odds ratios for suicidal ideation are large with wide confidence intervals due to the limited number of events. It would be helpful to further emphasize that these estimates should be interpreted as indicators of association rather than precise risk quantification.

Response:

We have revised the corresponding statement in the Discussion to further emphasize that the large odds ratios with wide confidence intervals should be interpreted as indicators of association rather than precise risk quantification, particularly given the limited number of suicidal ideation events.

Comment 5:

Please clarify that suicidal ideation represents a marker of psychological distress and potential risk, but does not equate to suicide attempts or suicide mortality.

Response:

We have added a statement in the Discussion to emphasize that suicidal ideation represents a marker of psychological distress and potential risk, but does not equate to suicide attempts or suicide mortality, and should be interpreted accordingly.

Comment 6:

The conclusion appropriately calls for integrated mental health screening. Adding one brief, concrete example (e.g., routine screening questions or referral pathways) may further enhance clinical relevance for non-psychiatric readers.

Response:

We have revised the Conclusion to include a brief, concrete example of how integrated mental health screening could be implemented in routine rheumatology practice, such as the use of brief screening questions during clinical visits and referral to mental health services when indicated, to enhance clinical relevance for non-psychiatric readers.

Reviewer 2:

This is an excellent and timely study that is highly relevant to contemporary Primary Health Care. The manuscript is logically structured, demonstrating high standards of academic writing and a clear understanding of the subject matter.

Section-by-Section Analysis

Introduction: The theoretical framework is sound. It effectively contextualizes the research problem and establishes a compelling rationale for the study.

Methodology: The research design is articulated with precision. The methodology is well-explained, providing sufficient detail for reproducibility and justifying the chosen approach.

Discussion: The interpretation of the findings is generally well-executed, situating the results within the broader context of existing Primary Health Care research. Limitations: The authors have shown commendable reflexivity. The study limitations are explicitly "enlightened" and transparently discussed, which strengthens the integrity of the findings.

Response:

We sincerely thank the reviewer for the positive and encouraging evaluation of our manuscript. We appreciate the recognition of the study’s relevance, methodological rigor, and clarity of presentation. We are also grateful for the reviewer’s acknowledgment of the theoretical framework, transparency in reporting, and the balanced discussion of limitations.

Editorial comments:

Comment 1:

Response:

We thank the editorial team for this reminder. We confirm that the manuscript has been prepared in accordance with PLOS ONE’s style and formatting requirements, including file naming conventions, and follows the official PLOS ONE templates for the main body as well as the title, authors, and affiliations.

Comment 2:

When completing the data availability statement of the submission form, you indicated that you will make your data available on acceptance. We strongly recommend all authors decide on a data sharing plan before acceptance, as the process can be lengthy and hold up publication timelines. Please note that, though access restrictions are acceptable now, your entire data will need to be made freely accessible if your manuscript is accepted for publication. This policy applies to all data except where public deposition would breach compliance with the protocol approved by your research ethics board. If you are unable to adhere to our open data policy, please kindly revise your statement to explain your reasoning and we will seek the editor's input on an exemption. Please be assured that, once you have provided your new statement, the assessment of your exemption will not hold up the peer review process.

Response:

We would like to confirm that all datasets generated and/or analysed during the current study have already been deposited in a public data repository and are freely accessible. The data are publicly available via the Harvard Dataverse at https://doi.org/10.7910/DVN/KCQKHM, in full compliance with PLOS ONE’s open data policy. Accordingly, no access restrictions apply, and no exemption is required.

Comment 3:

We notice that your Figure S2 is uploaded with the file type 'Figure'. Please amend the file type to 'Supporting Information'. Please ensure that each Supporting Information file has a legend listed in the manuscript after the references list.

Response:

We have amended the file type of Figure S2 to Supporting Information and ensured that the corresponding legend is listed in the manuscript after the references, in accordance with PLOS ONE guidelines.

Comment 4:

Add the value of Cronbach's alpha

Response:

We would like to clarify that the Cronbach’s alpha values for all psychometric scales used in the study have already been fully reported in the Supporting Information (Table S1: Preliminary validation metrics of psychometric scales in the study cohort). This table provides Cronbach’s alpha coefficients for PHQ-9, HAM-A, ISI, ASEX, and EQ-5D-5L, along with additional reliability and validity indicators.

Comment 5:

Type the references according to the PLoS ONE

Response:

We would like to confirm that all references in the manuscript have been formatted in accordance with the PLOS ONE reference style guidelines.

Comment 6:

Improve the quality of the figures

Response:

We confirm that all figures have been prepared and uploaded in accordance with PLOS ONE’s technical requirements, including resolution, format, and overall figure quality.

---

## [Editor Report · Decision Letter 1]

30 Jan 2026

SUICIDAL IDEATION, MENTAL HEALTH DISTURBANCES, AND QUALITY OF LIFE IN RHEUMATOID ARTHRITIS: EVIDENCE FROM A VIETNAMESE COHORT

PONE-D-25-66452R1

Ha Thi Thu Tran

Dear Dr. Ha Thi Thu Tran,

We’re pleased to inform you that your manuscript has been judged scientifically suitable for publication and will be formally accepted for publication once it meets all outstanding technical requirements.

Kind regards,

Marwan Salih Al-Nimer, MD, PhD

Academic Editor

PLOS One
---

## [Editor Report · Acceptance letter]

PONE-D-25-66452R1

PLOS One

Dear Dr. Tran,

I'm pleased to inform you that your manuscript has been deemed suitable for publication in PLOS One. Congratulations! Your manuscript is now being handed over to our production team.

Kind regards,

on behalf of

Professor Marwan Salih Al-Nimer

Academic Editor

PLOS One